# Why Are Farmers Reluctant to Sell: Evidence from Rural China

**Pan Wang and Di Liu ***

College of Economics and Management, Huazhong Agricultural University, Wuhan 430070, China
* Correspondence: ld94@webmail.hzau.edu.cn

**Abstract:** The agricultural marketing behavior of farmers is crucial for the realization of production value. Based on survey data from 406 citrus farmers in Hubei Province, this paper empirically examines the effects of risk aversion and Internet use on farmers' marketing behavior in terms of fresh produce. The results show that, first, farmers are generally reluctant to sell, with reluctant sellers accounting for about one-third of the total sample, and most report high levels of Internet use. Second, risk aversion and Internet use have a significant impact on farmers' reluctance to sell. The higher the level of farmer risk aversion, the lower the reluctance to sell, while Internet use significantly increases the probability of farmers being reluctant to sell. Third, Internet use weakens the inhibiting effect of risk aversion on reluctance to sell. These findings help to clarify the factors influencing farmers' reluctance to sell and provide reference suggestions for promoting high-quality agricultural development and rural industrial revitalization.

**Keywords:** risk avoidance; Internet use; reluctant to sell; citrus growers





## 1. Introduction

China is one of the largest producers of fruit in the world. According to the latest data, China's orchards cover 12,646.28 hm$^2$, and total fruit production has reached 286,923.6 kt [1]. However, price fluctuations in the Chinese fruit market are very frequent. Since 2009, the monthly fluctuation of fruit prices in 36 major cities in China has exceeded 5.01%, peaking at 14.51% [2]. Meanwhile, the uncertainty of agricultural activities has increased dramatically due to frequent emergencies, such as extreme weather conditions and COVID-19, increasing the severity of the fruit market price risks faced by farmers.

Many researchers have studied farmers' selling strategies when market prices fluctuate. They have found that farmers' sales behavior often shows a strange phenomenon known as "reluctance to sell" when faced with market price fluctuations [3]. "Reluctance to sell" first appeared in relation to grain purchases. It refers to the fact that farmers are not satisfied with the price of agricultural products or expect the price to increase, which leads to wait-and-see behavior or a reluctance to sell [4]. "Reluctance to sell" is a subjective and rational behavior of farmers, i.e., farmers can sell but do not sell, rather than having difficulty selling. For small-scale farmers in developing countries, profit maximization and risk minimization are their primary goals [5]. Although reluctance to sell may reduce losses or increase revenue, it is often accompanied by greater risks. This seems to be inconsistent with the view that "farmers are rational people".

Most studies focus on farmers' reluctance to sell stored agricultural products such as grains. For example, Qi and Yu [6] analyzed the factors influencing farmers' reluctance to sell. Mattos and Fryza [7] found that risk appetite contributed to Canadian wheat farmers' reluctance-to-sell behavior. Using an experimental approach, Vollmer et al. [8] similarly demonstrated the risk appetite of farmers' reluctance to sell in relation to grain. Luo et al. [9] analyzed the impacts of storage losses and market development on the maize-selling behaviors of rural households in China. Peng and Xu [10] found that large-scale farmers who anticipate an increase in grain prices will choose to postpone grain

sales to potentially maximize profits. Risk-averse large-scale farmers will choose a multiperiod sales method, while risk-loving ones will choose a single-period delayed sales method. Additionally, there have been fewer studies on fresh agricultural products such as fruits. The optimal time to sell agricultural products may vary depending on the product type. Fresh produce is perishable and less easily transported than stored agricultural products. Sun et al. [11] argued that the higher the expected price and the greater the market risk perception, the more growers prefer the long-term storage of high-value agricultural products.

In summary, although scholars have carried out numerous studies on farmers' reluctance to sell, there is room for further advancement. First, most existing studies focus on explaining the influencing factors and phenomena of reluctance to sell but ignore the internal relations between the factors and lack a comprehensive and systematic analysis. Second, scholars mostly focus on the sales behavior of large-scale farmers, and there is less research on small farmers. The current research on the selling behaviors of farmers in developing countries largely revolves around small-scale farmers and the impact of their endowments, ignoring the roles played by information technology and farmers' subjective attitudes.

This study takes risk aversion and Internet use as its perspectives. It uses survey data gathered from farmers in the main citrus-producing areas of Hubei Province to empirically investigate how smallholder farmers in China determine the optimal time to sell in the context of information asymmetry and market price fluctuations, i.e., whether farmers exhibit reluctant selling behavior. Ultimately, our goal is to understand the selling behavior of fruit farmers in developing countries such as China, to help smallholders increase their income, and to provide recommendations for promoting quality agricultural development and rural revitalization.

Compared with previous studies, the innovation and contributions in this study include the following. First, this study reveals the special sales behavior of farmers, which differs from the definition of farmers' sales behavior in previous studies. This study mainly focuses on farmers' reluctant selling behaviors. Second, this study combines the actual scenario of China's fresh fruit market to construct an analysis framework for small-scale farmers' sales decisions. Simultaneously, based on prospect theory and information effect, this study reveals the trade-off relationship between profit and risk in small-scale farmers' sales decision-making processes and the importance of information acquisition, which helps to understand the characteristics of their sales behavior.

## 2. Theoretical Framework

### 2.1. Farmers' Agricultural Products Sales Decision

Suppose that each rural household decides the optimal timing for selling agricultural products based on three main factors: proportional transaction costs, $TC^P$; fixed transaction costs, $TC^f$; and expected market price, $p'_j$. As the number of agricultural products sold increases, the proportional transaction costs, $TC^p_j$, faced by the farmer in period $j$ also increase. It depends on the market distance, $d_j$, trading time, $t_j$ and other characteristic factors, $z^p_j$, expressed as follows:

$$TC^p_j = TC^p\left(d_j, t_j, z^p_j\right) \tag{1}$$

The fixed transaction costs are not related to the number of sales, but to the fixed cost of each family, $z^f_j$, which can be expressed as follows:

$$TC^f_{ij} = TC^f\left(z^f_j\right) \tag{2}$$

In particular, due to the particularity of citrus planting, the production costs of citrus fertilizers and pesticides can also be regarded as fixed transaction costs each year. In our survey practice, we also found that farmers usually ignore the factor inputs in the citrus-growing process when deciding when to sell, as it is a perennial plant and can bear fruit multiple times. In contrast, farmers are more concerned with labor prices, storage costs and other transaction costs that vary with the quantity sold. We include production costs in our discussion of fixed transaction costs.

Farmers usually make sales decisions based on market prices, $p_j'$, $p_j'$, related to market price information, $\bar{p}_j$, and price fluctuations, $M.M$, which depend on the expected sales volume, $s_j$, the quality of agricultural products, $q_j$, and the bargaining power of farmers, $BP_j$. The function form of $p_j'$ can be expressed as follows:

$$p_j' = \bar{p}_j \pm M(s_j, q_j, BP_j) \tag{3}$$

Because agricultural production is susceptible to various factors, such as the external environment, it is characterized by typical uncertainty. Therefore, by adding risk variables, $\lambda$, the final sales volume of farmers in the current period is $(s_j - \lambda_j g)$, where $\lambda_j g$ represents the random yield loss caused by uncertain risk factors in period $j$. According to the principle of profit maximization, the optimal sales decision conditions of rational smallholders in period $j$ are as follows:

$$\max_j \Pi_{ij} = \left\{ (s_j - \lambda_j g) \cdot \left[ \bar{p}_j \pm M(s_j, q_j, BP_j) - TC_j^p \left( d_j, t_j, z_j^p \right) \right] - TC_j^f \left( z_j^f \right) \right\} \tag{4}$$

When the current sales profits reach the maximum, farmers will choose to sell agricultural products. It can be seen that proportional transaction costs, fixed transaction costs, market prices and risk factors have a key impact on agricultural product sales decisions.

### 2.2. Sales Decision Framework for Small-Scale Fruit Grower

When small-scale farmers, who aim to maximize profits and minimize risk, face exogenous price shocks under information asymmetry, their reluctance to sell is manifested. Additionally, subsequent decisions are made taking into account their price expectations and risk-averse attitudes. Prospect theory suggests that people make decisions based on reference points; individuals are willing to take more risk when faced with losses and tend to be risk-averse when faced with gains. Individuals are more sensitive to losses than gains [12].

Farmers face multiple risks, such as natural disasters and price fluctuations, during agricultural production and operation. Individual risk preferences will directly impact their behavioral decisions [13]. According to behavioral bias theory, individuals are more likely to maintain the status quo and avoid risk when faced with a choice between benefits and risks [14]. A reluctance to sell may reduce losses or increase revenues. However, it is often associated with greater risks. Specifically, on the one hand, farmers may hesitate for too long and lose their initial gains due to a fall in prices. On the other hand, the large number of homogeneous agricultural products on the market during the waiting period will further crowd out the market space and lead to stagnating sales of agricultural products, resulting in greater losses. In addition, scholars have found typical heterogeneity in the risk preferences of farm households [15,16]. As smallholder farmers have a minimal risk tolerance, most tend to be risk-averse to reducing economic losses. Therefore, risk-averse farmers who are averse to losses will choose to sell their agricultural products promptly and are more likely to be willing to sell to avoid losses or suffer greater losses caused by risk factors such as price fluctuations and natural disasters. Based on this, we propose the following hypothesis:

**H1:** *The more risk-averse small-scale farmers are, the lower the possibility of reluctant sales of their produce.*

Price expectations are another key factor affecting farmers' marketing strategies. Price expectations refer to farmers' prediction of future market price level trends. Since small-scale farmers are generally less educated, they judge the future price trend of agricultural products through experience or other information obtained. The ability to make correct price expectations will determine whether small-scale fruit growers can successfully maximize profits.

According to basic economic theory, a lack of information can lead to a loss of economic efficiency [17]. However, most rural areas are remote and have poor communication with markets, and farmers often need more accurate information on current market prices to make their decisions. The advent of Internet technology has broken down information barriers and influenced farmer behavior through the effects of information and technology [18].

On the one hand, Internet use can enhance the flow of information and reduce the financial and time costs required for farmers to obtain and exchange information, thus reducing transaction costs [19]. Additionally, abundant information flow can help farmers judge market trends and increase their market participation [20,21]. On the other hand, the Internet innovation platform can facilitate the production-marketing interface, which broadens the marketing channels of farmers and increases the sales of agricultural products [22]. In particular, the application of e-commerce has shortened the sales chain and greatly increased the market participation of farmers.

Furthermore, Internet use will enhance the factor allocation capacity of farmers and alleviate factor endowment constraints [23], thus optimizing farmers' marketing strategies related to agricultural products. The more farmers are in contact with outside information through the Internet, the more their sales thinking and approach will change, ultimately leading to changes in farmers' sales behavior. Generally, the more information a farmer has about the market, the more they expect to sell the product at a higher price. Internet use reduces the cost of trading agricultural products and expands trade channels. Therefore, when faced with price fluctuations, farmers with high Internet use will likely be reluctant to sell. Therefore, we propose the following hypothesis:

**H2:** *Farmers are more likely to be reluctant to sell their produce if they have a high level of Internet use.*

Chinese farmers have long had a weak position in the sales market, with intermediaries forming a monopoly market for small-scale farmers [24]. This is mainly because farmers know very little about market prices compared to intermediaries. However, Internet use can reduce the market power of intermediaries to some extent, improve the bargaining power of farmers, and thus increase the selling price of agricultural products [25]. Theoretically, Internet use mitigates the degree of risk aversion among farmers to a certain extent. The positive influence on the risk attitude of farmers who use the Internet more frequently is more significant [26,27].

Specifically, first, the change in information access channels brought about through the popularization of Internet technology in rural areas mitigates the degree of risk aversion of farmers through the effects of information [27]. Second, farmers' mastery of Internet technology can spill over to the level of knowledge network and social network. Better use of Internet technology can be internalized into farmers' knowledge endowment and social capital, improving farmers' ability to resist risks. Third, the Internet has improved farmers' access to credit [28]. Internet use can reduce farmers' liquidity constraints, mitigate the financial vulnerability of rural households, and encourage farmers to be more likely to engage in risky behavior.

In summary, the impact of Internet use on farmers' agricultural marketing behavior is in essence achieved by alleviating farmers' risk-averse attitudes, increasing farmers' risk resistance, and reducing rural households' liquidity constraints. The higher the risk appetite of farmers, the higher their resistance to risk, and the weaker their household liquidity constraints, the more likely they are to be reluctant to sell their produce. Given this, we propose the following hypothesis:

**H3:** *Internet use will mitigate the disincentive effect of risk aversion on farmers' reluctant selling behavior.*

## 3. Data

### 3.1. Data Collection

The research data were taken from a household survey of citrus growers in Hubei Province, China, conducted in July 2021. These survey areas were selected for two reasons: First, Yichang City, Hubei Province, as one of the important production areas of citrus in China, has rich reserves of varieties and a long and experienced history of cultivation. Second, as the most developed citrus industry in Yichang [29], Zigui County can basically reflect the current situation of citrus development in Yichang, and the study of its farmers' behavior is also generalizable to other production areas. Therefore, selecting the sample of citrus growers in Hubei Province is not only scientifically representative, but also has important practical value for the development of the citrus industry.

The survey first identified Zigui County, Yichang City, Hubei Province, as a sample area, and through a typical sampling method, selected two better citrus-producing towns, Guojiaba Town and Shuitianba Township. Secondly, the method of stratified random sampling was adopted, and the administrative villages were randomly selected according to the total annual output of citrus. A total of seven sample villages were formed. Finally, 60 citrus growers were randomly selected from each sample village for investigation. It is worth mentioning that the list of villagers provided by village cadres helps us better adapt to random selection. In each sample village, we ranked the citrus size from largest to smallest, and each sample farmer was selected to be distinguished by the same number of non-sample farmers. Finally, we carefully reviewed the survey data according to the integrity, logic, and authenticity of the questionnaire, obtaining 406 pieces of valid survey data for empirical analysis.

The interviewers consisted of doctoral and master's students from the research team, who had received advanced training. The interviewees were the main laborers or the heads of households responsible for citrus production and operation. Based on a randomly selected list, the interviewers entered the farmers' homes one by one for face-to-face communication. Then, the interviewers asked questions and filled out the questionnaire accurately according to the interviewees' answers. The main contents of the questionnaire included the input and output of citrus production, production and management decisions, and basic family and personal information.

### 3.2. Basic Characterization of the Sample

Table 1 reports the basic characteristics of the samples. Citrus marketing methods can be divided into two categories: only selling through intermediaries, and selling through intermediaries while also selling on their own (hereinafter referred to as "simultaneous self-selling"). The survey found that most farmers in the survey area rely on the sales method of 'farmers + middlemen', which is consistent with the research conclusions of Song and Qi [29]. However, while relying on intermediaries for sales, farmers also use other methods to sell on their own. Among all the survey samples, 141 households sell through online social networking platforms, accounting for 34.73% of the total sample. It can be seen that the Internet sales model based on social networking platforms is gradually being integrated into the farmers' sales process.

**Table 1.** The distribution of the sample.

| Variable | Description | Observation | Percentage | Variable | Description | Observation | Percentage |
|---|---|---|---|---|---|---|---|
| Gender | Male | 227 | 44.09 | Education | Primary and below | 204 | 50.25 |
| | Female | 179 | 55.91 | | Junior high | 147 | 36.21 |
| Age | <45 | 64 | 15.76 | | Senior High School | 46 | 11.33 |
| | 45~60 | 224 | 55.17 | | Junior college | 7 | 1.72 |
| | >60 | 118 | 29.06 | | Bachelor and above | 2 | 0.49 |
| Orchard area (mu) | <4 | 120 | 29.56 | Sales methods | Intermediaries | 237 | 58.37 |
| | 4~8 | 207 | 50.99 | | Multiple channels | 169 | 41.63 |
| | >8 | 79 | 19.46 | | E-commerce platforms | 23 | 5.67 |
| Citrus sales revenue | <50,000 | 223 | 54.93 | | Social platform | 141 | 34.73 |
| | 50,000~10,0000 | 137 | 33.74 | Multiple channels | Short video platform | 11 | 2.71 |
| | 100,000~150,000 | 33 | 8.13 | | Supply to supermarkets | 6 | 1.48 |
| | ≥150,000 | 13 | 3.2 | | Sell offline to customers | 36 | 8.87 |

Note: While relying on intermediaries for sales, farmers also use other methods to sell on their own; the way mainly includes 5 kinds: e-commerce platform, social platform, short video platform, supply to supermarkets and sell offline to customers. 1 mu = 1/15 hectare.

## 4. Empirical Model and Descriptive Statistics

### 4.1. Empirical Model

We construct a binary Probit model of citrus growers' sales behavior choice for analysis. The expression is as follows:

$$Y = \ln\left(\frac{p}{1-p}\right) = \beta_0 + \beta_1\varphi + \beta_2\eta + \beta_3 X + \mu \tag{5}$$

where $p$ represents the probability that citrus growers adopt reluctant selling behavior; $1 - p$ represents the probability that farmers sell normally. $\varphi$ and $\eta$ represent the degree of risk aversion and the level of Internet use of farmers, respectively. $\beta_1$ and $\beta_2$ are their estimated coefficients, respectively. $X$ is the control variable vector that affects the decision-making processes of farmers' sales behavior, and $\beta_3$ is its estimated coefficient vector. $\beta_0$ is a constant term and $\mu$ is a random error term.

In order to further explore whether the level of Internet use of farmers will alleviate the inhibitory effect of risk aversion on farmers' reluctance to sell, we add the interaction term of $\varphi$ and $\eta$ on the basis of Equation (5):

$$Y = \ln\left(\frac{p}{1-p}\right) = c_0 + c_1\varphi + c_2\eta + c_3(\varphi \times \eta) + c_4 X + \omega \tag{6}$$

In Equation (6), $\varphi \times \eta$ is the interaction term between farmers' risk aversion and Internet use level, and $c_3$ is its estimated coefficient. $c_1$ and $c_2$ are the estimated coefficients of $\varphi$ and $\eta$, respectively. $X$ is the vector of control variables, and $c_4$ is its estimated coefficient. $c_0$ is a constant term, and $\omega$ is a random error term.

### 4.2. Key Variables and Descriptive Statistics

In order to estimate the coefficients mentioned earlier, we need to define the variables. For the dependent variable, we selected 'whether citrus growers take reluctant selling behavior' in the questionnaire item to identify Y. Additionally, this study included two key independent variables. One of them is risk aversion. As the more risk-averse farmers tend to delay the adoption of new technologies [30,31], we refer to the research of Wang et al. [32], through the following question: 'Are you willing to try the new citrus planting technology' for identification. This is measured using a scale of 1 to 5, with lower values indicating

that farmers are more risk-averse and higher values indicating that farmers show a greater preference for risk.

The other is Internet use. We measure the Internet use level of citrus growers from five levels: learning, work, social, entertainment and business activities. These five categories of indicators are assigned from 1 to 7 according to the frequency of Internet use. In ascending order of fetching value, they represent seven levels: never, once every few months, once a month, 2–3 times a month, 1–2 times a week, 3–4 times a week, and almost every day. We use the mean score of these indicators to identify the Internet use of citrus growers. A higher score for farmers represents a higher level of Internet use. It is worth mentioning that compared with previous studies, we pay more attention to the farmer's Internet skills acquisition rather than simply measuring whether farmers can use the Internet.

Of course, we also controlled for the impacts of other factors on farmers' citrus sales behavior in this study. On the one hand, drawing on previous studies, we controlled the personal, family and production characteristic variables of citrus growers. Personal characteristics include gender, age, and education; family characteristics include cooperative membership, labor force and economic status; and production characteristics include planting scale, yield, planting years, the proportion of sales revenue and cooperative relationships. On the other hand, the transportation condition and market distance at the village level are controlled. The definition and characteristic statistical results of each variable in the model are shown in Table 2.

**Table 2.** Variable definitions and descriptive statistics.

| Variables | Definition and Assignment | Mean | Standard Deviation |
|---|---|---|---|
| | Dependent variable | | |
| Citrus sales behavior | 1 if the farmer is reluctant to sell, otherwise 0 | 0.320 | 0.467 |
| | Key independent variables | | |
| Risk aversion | New technology adoption intention: Likert 1–5 points, 1 = very unwilling, 5 = very willing | 2.773 | 1.032 |
| Internet use | Means of 5 categories of indicators | 3.607 | 1.340 |
| Gender | 1 = Male, 0 = Female | 0.559 | 0.497 |
| Age | Citrus growers' actual age(years) | 54.892 | 10.589 |
| Education | 1 = Primary and below; 2 = Junior high; 3 = Senior high school; 4 = Junior college; 5 = Bachelor and above | 1.660 | 0.784 |
| Cooperative membership | 1 if farmer is a cooperative member,0 otherwise | 0.197 | 0.398 |
| Labor force | Total household labor force (person) | 3.032 | 1.094 |
| Economic status | 1 = Very poor, 2 = Poor, 3 = Medium, 4 = Good, 5 = Very good | 2.640 | 1.030 |
| Planting scale | Actual citrus planting area (mu) | 7.142 | 24.847 |
| Yield | Average yield per mu (kg), taking the natural logarithm | 7.958 | 0.417 |
| Planting years | Citrus planting duration(years) | 27.744 | 10.060 |
| Proportion of sales revenue | Total revenue from sales of citrus as a percentage of total family revenue (%) | 81.344 | 28.055 |
| Cooperative relationship. | 1 = Very poor, 2 = Poor, 3 = Medium, 4 = Good, 5 = Very good | 3.155 | 1.119 |
| | Village characteristics | | |
| Transportation condition | 1 = Poor; 2 = General; 3 = Better | 2.037 | 0.835 |
| Market distance | Distance to the nearest agricultural trading market in the village (km) | 7.850 | 9.539 |

Note: 1 mu = 1/15 hectare.

To check the suitability of the Internet use survey scales, the Kaiser–Meyer–Olkin (KMO) test and Bartlett's sphericity test are used. The Kaiser–Meyer–Olkin measure of sampling adequacy is 0.696, and the Bartlett test of sphericity is 718.745 ($p$ = 0.000), indicating that the survey scales are suitable for factor analysis. The results of the factor analysis show that the Internal consistency (Cronbach's $\alpha$) and composite reliability (CR) are greater than 0.7, and the average variance extracted (AVE) is greater than 0.5, indicating that the scale has good reliability and validity. The specific results are shown in Table 3.

**Table 3.** Description of Internet use.

| Variable | Subject | Mean | Std. | Standard Factor Loadings | Cronbach's $\alpha$ | CR | AVE |
|---|---|---|---|---|---|---|---|
| | Learning | 2.022 | 1.417 | 0.724 | | | |
| | Working | 2.163 | 1.635 | 0.777 | | | |
| Internet use | Social | 5.293 | 1.873 | 0.715 | 0.778 | 0.853 | 0.538 |
| | Entertainment | 5.488 | 2.027 | 0.678 | | | |
| | Business | 3.069 | 2.159 | 0.769 | | | |

Note: We use a 7-point Likert scale to measure Internet use, with scores of: 1 = never; 2 = once every few months; 3 = once a month; 4 = 2–3 times a month; 5 = 1–2 times a week; 6 = 3–4 times a week; 7 = almost every day.

In addition, Table 4 illustrates that 130 households out of the total sample of farmers exhibit reluctant selling behavior, accounting for 32.02% of the total sample. Among the sample farmers who adopt the reluctant sale behavior, the proportion of high level of Internet use is greater. The Pearson correlation coefficient between Internet use and sales behavior is statistically significant at the 1% level, indicating a significant positive relationship.

**Table 4.** Farmers' sales behavior under the difference of Internet use level.

| Variable | Sales Behavior | | | |
|---|---|---|---|---|
| | No Reluctant Selling | | Reluctant Selling | |
| | Observation | Percentage (%) | Observation | Percentage (%) |
| High level of Internet use | 146 | 76.40% | 45 | 23.60% |
| Low level of Internet use | 130 | 60.50% | 85 | 39.50% |
| Pearson correlation coefficient | | 0.171 *** | | |

Note: *** is significant at the 1% level.

## 5. Estimation Results and Robustness Test

### 5.1. Estimation Results

Table 5 reports the results of the benchmark regressions on the effects of risk aversion and Internet use on citrus growers' selling behavior decisions. Model 1 is the result of considering only the control variables. Model 2 and Model 3 are the results of risk aversion and Internet use each affecting farmers' marketing behavior decisions. Model 4 is the result of adding the interaction term of risk aversion and Internet use.

First, considering only the control variables, Model 1 shows that planting scale, yield, and proportion of sales revenue share significantly positively affect farmers' reluctant selling behavior. In contrast, the village transportation development and market distance have a significant negative effect. The possible reasons are as follows: First, farmers with larger citrus planting areas, higher yields and larger proportions of citrus sales revenue have a higher dependence on citrus sales revenue. They expect to sell at a higher price, so their wait-and-see time is longer, and the possibility of being reluctant to sell is higher when the market price fluctuates. Second, the higher the level of transportation development, the farmers can choose more sales channels through various ways to sell agricultural products promptly. Moreover, the closer the farmers are to the market, to save transaction costs, it is easier to choose to sell nearby, and farmers are less likely to make reluctant sales.

Second, according to the estimation results shown in Model 2, citrus growers' risk aversion has a significant negative impact on reluctant selling behavior after controlling for other influencing factors. Thus, Hypothesis 1 is supported. This is due to reluctant selling behavior greatly increasing the risks and costs that farmers need to bear, meaning that they easily miss the best time to sell [33]. However, risk-averse farmers are usually more cautious when making decisions [23], so they are less likely to be reluctant to sell.

**Table 5.** Benchmark regression results.

| Variable | Model 1 | Model 2 | Model 3 | Model 4 |
|---|---|---|---|---|
| Risk aversion | — | −0.223 *** (0.073) | — | −0.292 *** (0.078) |
| Internet use | — | — | 0.168 ** (0.066) | 0.202 *** (0.067) |
| Risk aversion × Internet use | — | — | — | 0.144 *** (0.054) |
| Gender | −0.037 (0.144) | −0.011 (0.146) | −0.039 (0.147) | 0.033 (0.152) |
| Age | 0.001 (0.009) | 0.002 (0.009) | 0.008 (0.009) | 0.009 (0.010) |
| Education | 0.086 (0.103) | 0.114 (0.105) | 0.006 (0.107) | 0.010 (0.111) |
| Cooperative membership | −0.040 (0.184) | −0.014 (0.187) | −0.110 (0.187) | −0.112 (0.194) |
| Labor force | −0.101 (0.067) | −0.108 (0.068) | −0.109 (0.067) | −0.104 (0.068) |
| Economic status | 0.027 (0.072) | 0.082 (0.075) | 0.009 (0.071) | 0.071 (0.075) |
| Planting scale | 0.043 ** (0.019) | 0.050 ** (0.021) | 0.035 * (0.019) | 0.039 * (0.020) |
| Yield | 0.447 ** (0.178) | 0.435 ** (0.182) | 0.408 ** (0.178) | 0.407 ** (0.182) |
| Planting years | 0.006 (0.008) | 0.004 (0.008) | 0.004 (0.008) | 0.002 (0.008) |
| Proportion of sales revenue | 0.010 *** (0.003) | 0.010 *** (0.003) | 0.010 *** (0.003) | 0.010 *** (0.003) |
| Cooperative relationship. | −0.028 (0.065) | −0.038 (0.065) | −0.018 (0.064) | −0.051 (0.066) |
| Transportation development | −0.542 *** (0.098) | −0.535 *** (0.099) | −0.558 *** (0.100) | −0.546 *** (0.102) |
| Market distance | −0.068 *** (0.010) | −0.070 *** (0.011) | −0.070 *** (0.010) | −0.071 *** (0.011) |
| Constant | −3.591 ** (1.619) | −3.039 * (1.641) | −3.987 ** (1.645) | −3.445 ** (1.677) |
| Observations | 406 | 406 | 406 | 406 |
| Wald chi2 | 78.930 *** | 80.000 *** | 81.230 *** | 89.330 *** |
| Pseudo r-squared | 0.199 | 0.216 | 0.211 | 0.242 |

Note: ***, ** and * are significant at the 1, 5 and 10% levels, respectively; standard errors are in parentheses.

Third, the estimation results in Model 3 show that Internet use significantly positively affects citrus growers' reluctant selling behavior. Hypothesis 2 has been verified. This suggests that even though farmers use the Internet to gain access to a wealth of market information, they choose to wait and see in terms of prices in order to maximize profits based on a full weighing of external information. Farmers tend to be reluctant to sell, consistent with the judgment that farmers demonstrate 'bounded rationality' in the prospect theory [34].

Finally, the results of Model 4 show that the estimated coefficient of risk aversion is negative, the estimated coefficient of Internet use is positive, and the estimated coefficient of the interaction between risk aversion and Internet use is positive, and both are significant at the level of 1%. This shows that Internet use can weaken the inhibitory effect of risk aversion on farmers' reluctance to sell, which verifies Hypothesis 3. The reason is that the higher the level of Internet use of farmers, the stronger the ability to obtain information, which helps to compensate for their information asymmetry in the market and reduces uncertainty in the sales process of agricultural products [19]. Whether farmers are reluctant to sell citrus is a typical uncertainty decision. Effective information acquisition reduces farmers' uncertainty perception [35], and then promotes them to adopt reluctant selling behavior.

### 5.2. Robustness Test

In the baseline regression model, we control for many variables that affect farmers' selling behavior. Additionally, there is also a lack of evidence for the theory that the risk attitude of farmers is affected by their sales behavior. Therefore, the possibility of endogenous problems caused by risk aversion and farmers' reluctant selling behavior in the model is not great. Even so, there may still be endogeneity problems due to omitted variables, and there may still be some unobservable variables that can have an impact on farmers' selling decisions. To overcome potential endogeneity problems, this study re-estimates the model by replacing the regression model, limiting the sample, and re-measuring the key variables.

Specifically, the Logit model is first used for re-estimation. Meanwhile, considering that the risk attitude of the elderly in rural areas is less likely to be changed by external factors, we refer to the study of [27], excluding the sample of farmers aged 60 and above

and then substituting the processed data into the original model for regression. Further, in order to verify the impact of key variables from multiple perspectives, this study also draws on Cai et al. [36] by assigning the first three options of the risk-aversion measure to 0 for "no" and the last two options to 1 for "yes", redefining it as "risk aversion".

Finally, "farmers' ability to grasp and receive Internet information" is used to characterize farmers' Internet use, with 1~5 representing "very poor, poor, medium, good and very good", respectively. The stronger the ability of farmers to grasp and receive Internet information, the higher the level of Internet use. Table 6 shows the results of various robustness testing methods. After controlling for endogeneity problems, the estimation results of the key variables are consistent with those shown in Table 5, indicating that the results are robust and credible.

**Table 6.** Robustness test results.

| Variables | Replacing the Regression Model | | Limiting the Sample | | Re-Measuring the Key Variables | |
|---|---|---|---|---|---|---|
| Risk aversion | −0.406 *** (0.131) | −0.502 *** (0.140) | −0.240 *** (0.091) | −0.347 *** (0.100) | −0.751 *** (0.233) | −1.167 *** (0.354) |
| Internet use | 0.306 *** (0.115) | 0.341 *** (0.119) | 0.198 ** (0.084) | 0.235 *** (0.086) | 0.314 *** (0.088) | 0.356 *** (0.093) |
| Risk aversion × Internet use | — | 0.237 ** (0.096) | — | 0.164 ** (0.068) | — | 0.515 ** (0.229) |
| Control variables | Yes | Yes | Yes | Yes | Yes | Yes |
| Observations | 406 | 406 | 272 | 272 | 272 | 272 |
| Wald chi2 | 74.150 *** | 78.208 *** | 71.610 *** | 75.140 *** | 76.080 *** | 73.370 *** |
| Pseudo r-squared | 0.233 | 0.244 | 0.254 | 0.268 | 0.276 | 0.289 |

Note: ***, ** are significant at the 1 and 5 levels, respectively; standard errors are in parentheses.

## 6. Conclusions, Policy Implications and Limitations

Determining the best time to sell fresh fruit is the most important economic decision for citrus growers. Based on the survey data of 406 citrus growers in China's major citrus-producing areas, we empirically analyzed the impact of risk aversion and Internet use on farmers' reluctance to sell. We used various methods to test the robustness to address the potential endogenous problems, and the estimated results were still valid.

Our study shows that: First, citrus growers generally have a reluctant selling mentality. Farmers reluctant to sell account for about one-third of the total number of the sample, most of whom have a high level of Internet use. Second, risk aversion and Internet use significantly impact citrus growers' reluctant selling behavior. The higher the risk aversion of farmers, the lower the possibility of being reluctant to sell, and Internet use can significantly increase farmers' reluctance to sell. Third, Internet use can weaken the disincentive effect of risk aversion on citrus growers' reluctance to sell.

Our findings also inform agricultural development in China and other developing countries. First, governments should gradually establish significant agricultural production, supply and demand, price, monitoring and an early warning system. It is necessary to use Internet technology fully, build an information platform, collect feedback in a timely manner, and convey agricultural production and management information. In particular, analyzing the impact of significant events on the market provides farmers with reliable information security, improving their ability to judge future market trends.

Second, governments should strengthen agricultural industrialization and create a community of interest. Considering the fragmented nature of the smallholder economy, governments should support and regulate the development of agricultural organizations such as cooperatives and associations to bring together the scattered smallholders. At the same time, the government should vigorously develop contract agriculture, establish effective market docking, improve the market competitiveness of farmers, and reduce their income risks.

Third, governments should improve the agricultural insurance system, build high-quality storage facilities and logistics systems, and establish a corresponding agricultural insurance protection mechanism for farmers according to their needs and enhance their risk-coping ability. At the same time, storage facilities should be established to reduce the loss of agricultural products.

Fourth, the important role of the Internet should be given full play in helping agricultural production and operation. Internet knowledge and skills training should be provided to farmers, and personalized training courses for farmers with different educational levels should be carried out to enhance the initiative of farmers to use Internet technology to optimize agricultural production and management decisions.

However, this study also has some limitations. Firstly, although our study area, Hubei Province, is the main citrus-producing area, other producing areas remain. Secondly, the sample size of our study is small, which will affect the robustness of the results to some extent. Finally, our study sample is limited to citrus growers, and there may be differences in the factors affecting the marketing of other types of agricultural products. Therefore, more future studies could be conducted in other citrus-producing areas in China, and the sample size could be expanded. Attention should also be paid to the factors influencing the reluctant selling behavior of other agricultural products.

**Author Contributions:** Conceptualization, P.W.; Data curation, P.W. and D.L.; Formal analysis, P.W.; Investigation, P.W. and D.L.; Methodology, P.W. and D.L.; Software, P.W.; Supervision, D.L.; Validation, P.W. and D.L.; Visualization, P.W.; Writing—original draft, P.W.; Writing—review and editing, D.L. All authors have read and agreed to the published version of the manuscript.

**Funding:** This research received no external funding.

**Institutional Review Board Statement:** Not applicable.

**Informed Consent Statement:** Informed consent was obtained from all subjects involved in the study.

**Data Availability Statement:** The data that are presented in this study are available from the corresponding author upon request. The data are not publicly available due to privacy restrictions.

**Conflicts of Interest:** The authors declare no conflict of interest.

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
