# Peer review of "Why Are Farmers Reluctant to Sell: Evidence from Rural China"

_agriculture, doi:10.3390/agriculture13040814_

Round 1

Reviewer 1 Report

I congratulate the authors for the topic, it is necessary, and more so for smallholder citrus farming, which has very similar problems in all parts of the world.

I would ask for a general revision of the text to polish some parts that seem redundant, or repetitive. The text is not always easy to understand, for example I see that the words risk, aversion, internet or reluctant are used repeatedly, I am sure you could look for some synonyms in order not to make the reading so heavy in some paragraphs.

In the title there is no reference to the use of the internet, which is the subject on which you have focused your work. Would you not consider changing the title to make it more realistic for the study?.

The term "reluctant to sell" is used too often, in reality no one who produces food for commercial purposes (I therefore exclude self-sufficiency agriculture) can afford to give up selling. Every farmer has to sell, sooner or later, otherwise there is no point in doing business. I think that what you often mean is that producers have a difficulty to sell, which is what actually happens in reality.

Always using the word "reluctant" gives the impression that not selling is a capricious attitude of the producer, a personal decision, an option, when it is always the consequence of their poor negotiating power, and even more so if they have small farms. This perceived image of the producer does not seem to me to be the most appropriate.

In paragraph 2.1 they refer to "transation costs", should they not refer to the term "production costs" at least some of the time? The producer is very limited by what it costs him to produce his fruit, then he must assume other costs derived from its sale (transport costs, logistics, etc.), but above all he must assume his "production costs", and these are not properly visualised in this section.

The wording of Hypothesis 1 could be improved, they use a double negation and the final meaning of the sentence is difficult to understand. You say: H1-The more risk-averse small-scale farmers are, the LESS LIKELY they are TO ADOPT RELUCTANT SELLING behaviour for their produce.

The tables can also be improved in their editing. In Table-1, the columns and rows get confused, note that you have two groups of data distributed in 4 columns, but the last row uses a total of 8 columns.

Table-5 is the same, surely you can put the two figures of each variable next to each other, and not one below the other, which makes it difficult to interpret. If this is not possible, put lines to separate the values between rows, although this option may be less visually appealing.

In table-3, we recommend that the values of the "measure" concept be listed as a table footer, not as a column, as this makes it difficult to understand the rest of the values.

In table-2 there is a figure that I do not understand. Given that all the points are decimals, how can it be that they manage harvests of 7.958 kilos per mu, it really seems to me to be a totally insufficient value if it is transferred to its value in kilos per hectare, is that figure correct?.

Finally, there are two errors in the quotations that you have included and which appear in rows 316 and 331.

Author Response

Dear Reviewer,

Thank you very much for giving us an opportunity to revise our manuscript. We appreciate editor and reviewers very much for their positive and constructive comments and suggestions on our manuscript entitled “Why are Farmers Reluctant to Sell: Evidence from Rural China” (ID: agriculture-2275610). Those comments are all valuable and very helpful for revising and improving our paper, as well as the important guiding significance to our researches. We have studied comments carefully and have made correction which we hope meet with approval. We use the revision mode "Track" in word. 
Please see the attachment.

Reviewer 2 Report

Title

I title of the article is adequate since it is reflective of its content.

 Abstract

I think the abstract is clear and concise enough for the reader to glean the main contribution of the study. However, I think the abstract could include a brief reference to the study’s contribution to theory and practice.

 Introduction

The introduction clearly describes the motivation of the study and adequately relate it to earlier work in the field. If follows a convincing and rational path of development, is very updated and covers a significant spectrum of extant empirical work.  The introduction also clearly defines the problem and identifies the research gap. Good work.

 Theoretical Framework

The theoretical framework is clearly described. The hypotheses are well sustained based on behavioral bias theory, economic theory and logical argumentation.

 Methodology

The methodology is solid. The criteria to select the sample was explained and the sample dimension is significative. The questionnaire process is transparent. The statistical procedures seem adequate, considering the objectives of the study.

 Results

The results of the study are well presented. The authors should however correct the references error in Pag. 8, line 316.

 Conclusions

Conclusions are well supported by the study.  However, the authors should include a reference to the limitations of the study.

Author Response

Dear Reviewer,

Thank you very much for giving us an opportunity to revise our manuscript. We appreciate editor and reviewers very much for their positive and constructive comments and suggestions on our manuscript entitled “Why are Farmers Reluctant to Sell: Evidence from Rural China” (ID: agriculture-2275610). Those comments are all valuable and very helpful for revising and improving our paper, as well as the important guiding significance to our researches. We have studied comments carefully and have made correction which we hope meet with approval. We use the revision mode "Track" in word. 

Round 2

Reviewer 1 Report

I congratulate you on the improvements you have made. I appreciate your effort in answering all the questions I have asked. I can understand that there are some things, for example, the change of title, that you do not do, although I still think that when reading the work it would always be more understandable.

I have one last doubt, and that is whether in line 251 when they say Equation (1) it is not Equation (5)?... Also, in line 252, instead of Equation (2), don't they mean Equation (6)?

All the best